# An Automatic Lie Detection Model Using EEG Signals Based on the Combination of Type 2 Fuzzy Sets and Deep Graph Convolutional Networks

**DOI:** 10.3390/s24113598

**Published:** 2024-06-03

**Authors:** Mahsan Rahmani, Fatemeh Mohajelin, Nastaran Khaleghi, Sobhan Sheykhivand, Sebelan Danishvar

**Affiliations:** 1Biomedical Engineering Department, Faculty of Electrical and Computer Engineering, University of Tabriz, Tabriz 51666-16471, Iran; mahsanrahmani@tabrizu.ac.ir (M.R.); nastrnkhaleghi@tabrizu.ac.ir (N.K.); 2Psychology Department, University of Aston, Birmangham B4 7ET, UK; 3Department of Biomedical Engineering, University of Bonab, Bonab 55517-61167, Iran; s.sheykhivand@tabrizu.ac.ir; 4College of Engineering, Design and Physical Sciences, Brunel University London, Uxbridge UB8 3PH, UK

**Keywords:** CNN, EEG, deep learning networks, lie detection

## Abstract

In recent decades, many different governmental and nongovernmental organizations have used lie detection for various purposes, including ensuring the honesty of criminal confessions. As a result, this diagnosis is evaluated with a polygraph machine. However, the polygraph instrument has limitations and needs to be more reliable. This study introduces a new model for detecting lies using electroencephalogram (EEG) signals. An EEG database of 20 study participants was created to accomplish this goal. This study also used a six-layer graph convolutional network and type 2 fuzzy (TF-2) sets for feature selection/extraction and automatic classification. The classification results show that the proposed deep model effectively distinguishes between truths and lies. As a result, even in a noisy environment (SNR = 0 dB), the classification accuracy remains above 90%. The proposed strategy outperforms current research and algorithms. Its superior performance makes it suitable for a wide range of practical applications.

## 1. Introduction

In recent decades, truth detection and lie detection tests have attracted the attention of many enthusiasts due to the increase in security threats and crime prevention and control. Many efforts have been made to design effective lie detection systems, and thus, advanced neuroscience-based methods for behavioral research have piqued the interest of scientists and researchers [1].

The most popular technique for detecting confirmation of hidden information is the polygraph. This approach is predicated on the idea that lying can cause various physiological reactions that can be seen and documented with the right equipment. Physiological responses are used in the polygraph to study the body’s involuntary alterations [2]. A polygraph assesses involuntary body changes such as skin conductance, heart rate, blood pressure, and breaths per minute [3]. To determine the subject’s level of honesty, the operator of the polygraph machine compares the measured physiological values to the expected normal levels of physiological signals following the test. However, despite its good performance, the polygraph is untrustworthy because experienced criminals can maintain normal physiological functions while being interrogated by the examiner with a polygraph and deceive both the examiner and the polygraph machine. As a result, the polygraph test results are not legal or valid [4]. However, in the recent decade, technologies beyond the polygraph, such as brain signals or electroencephalogram (EEG), have been created to identify truths and lies [5,6] accurately. EEG waves can help discriminate between truth and lies. EEG is employed in various medical applications in patients, such as brain–computer interface (BCI) and epilepsy diagnosis [6]. Brain signals are among the human electrical signals. Nerve cells in the brain produce electrical impulses that change in distinct wave patterns regularly [7]. EEG is the recording of electrical activity on the head using electrodes. Electroencephalography can identify lying by analyzing aberrant brain wave variations. These signals are challenging to classify because of their instability and low signal-to-noise ratio (SNR) [8]. After recording the signal, the primary purpose is to interpret, analyze, and transform the waves into a human-readable format for input for various devices. For this purpose, recent years have seen the development of research into the creation of lie detection systems based on EEG, which is discussed below.

Abutalebi and colleagues [9] studied the extraction of EEG features in P300-based lie detection. As a result, these researchers developed a novel technique based on specific features and statistical classification. In this study, the researchers used Ag/AgCl electrodes in the Fz (frontal area), Cz (central area), and Pz (parietal lobe) locations of the 10–20 system to record EEG signals at a sampling rate of 256 Hz. The best features in this study were determined as input feature vectors for the classifier using a genetic algorithm (GA). The researchers chose morphological, frequency, and time series features. According to this study, the rate of correct diagnosis based on the two classes of guilty and innocent is as high as 86%. Amir and colleagues [10] investigated lie detection using EEG signal processing during interrogations. In this study, frequency bands of brain waves were first extracted. The second step involved extracting morphological features such as amplitude, peak, and delay from existing waves. This study used a standard 10–20 system to record five channels of EEG signals. The study concluded that increasing the number of electrodes in the signal recording yielded more accurate results for distinguishing truth from lies. Mohammad and colleagues [11] investigated how human emotions change while lying using EEG and electrooculography (EOG) signals. This study had ten participants ranging in age from 18 to 28. EEG electrodes were applied to the patient’s scalp using a standard 10–20 system with 32 channels. Furthermore, the sampling rate used to record the signal for each channel was 2000 samples per second. In this study, the delta waves in the supine position had the greatest effect on separation, resulting in a classification accuracy of 67%. Furthermore, the remaining theta, alpha, beta, and gamma waves had maximum accuracy of 52.15%, 55.10%, 79.6%, and 13%, respectively. In this study, the researchers determined that electroencephalography is an accurate and sensitive method for measuring emotional expression while lying. Gao and colleagues [12] surveyed P300-based lie detection techniques. They developed a new method to improve the SNR ratio of the P300 wave, which is used to increase the accuracy of separating lies from truth. In this study, 14 EEG channels from 34 patients were recorded. The P300 wave with a high signal-to-noise ratio was obtained using a new spatial denoising method based on independent component analysis (ICA). This study extracted features in the time domain as well as the frequency domain. This study used the support vector machine (SVM) classifier to classify the feature vector. The maximum accuracy obtained in this study was reported to be 96%. Simbolon and colleagues [13] presented an intelligent system for lie detection based on EEG signals using an SVM classifier. They used Fz, Cz, Pz, O1, and O2 channels to record the signal. The features used in this study were mean, standard deviation, median, maximum, and minimum. The researchers reported a final accuracy of around 70%. Although the classification accuracy was low in this study, it could distinguish between all classes (both false and true). The study’s second advantage is the use of minimal signal-recording electrodes. Saini and colleagues [14] investigated the classification of EEG signals using various features for lie detection. This paper described a novel approach to extracting and integrating domain features with an SVM classifier. EEG data were collected using the international 10–20 electrode placement system, which consisted of channels C3, C4, P3, Pz, P4, O1, O2, and Oz. The Pz channel produced the best results in the analysis of recorded electrodes. This study employed time, frequency, wavelet transform (WT), and empirical mode decomposition (EMD) parameters. Finally, 40 features were extracted from the data and classified with an SVM classifier. The researchers reported a maximum accuracy rate of 98%. Despite the high accuracy in separating the classes, this research has a high computational volume and is not suitable for use in real-time systems. Yohan and colleagues [15] proposed a lie detection system that used EEG signals from SVM, K-nearest neighbor (KNN), artificial neural networks (ANNs), and linear classifiers (LRs). The recorded signal was processed with a fast Fourier transform (FFT) to extract features. Among the classifiers tested, the SVM classifier had the highest accuracy (86%) for classifying lie and truth. Bagel and colleagues [16] used deep convolutional networks to distinguish between truth and lies based on EEG data automatically. Their research aimed to develop a deep learning-based model capable of distinguishing truth from lies while not controlling emotions or physiological expressions. The proposed model was trained and validated using the DRYAD dataset. In this dataset, 30 people were randomly assigned to the guilty and innocent groups, and the stimulus was evaluated while brain signals were recorded. These researchers proposed a network in which low-level features were extracted for the first layers. Furthermore, their proposed network had varying numbers of neurons and modified rectified linear unit (ReLU), hyperbolic tangent, and sigmoid activation functions. The accuracy reported for classification using the proposed method by these researchers was 84%. Dodia and colleagues [17] suggested an Extreme Learning Machines (ELMs)-based lie detection system using EEG signals. The researchers recorded the EEG signal using 16 Ag/AgCl electrodes. In their study, the recorded signal was first preprocessed to eliminate noise. The signal was then analyzed using algorithms such as the Fourier transform (FT) to extract features. The researchers’ study identified features such as mean, variance, maximum, minimum, skewness, elongation, and power. Finally, the feature vector was classified with the ELM classifier. The maximum reported accuracy for the classification proposed by these researchers was 88%. Kang and colleagues [4] created a lie detection system using deep learning. This study employed independent component analysis (ICA) and clustering techniques. In addition, this study used a functional connection network (FCN) classifier to classify the lie and truth classes. This study discovered that lying improves information exchange between the frontal and temporal lobes. The final accuracy reported in this study was 88%. Boddu and colleagues [6] demonstrated a lie detection system based on EEG signals. This study enhanced EEG channels using the particle swarm optimization (PSO) algorithm. Based on this, only PSO-selected channels were used in the study. The proposed approach in this study, which is based on SVM classification, achieved an accuracy of 96%. The classifier’s high accuracy was one of the study’s advantages; however, one of its limitations was the use of class feature extraction and selection.

A review of previous studies on the automatic detection of truth from lies using EEG signals reveals that, while many studies have been conducted in this field, there are still numerous limitations. These limitations and challenges are thoroughly examined below: (A) All prior research (apart from a single instance) retrieved the feature vector from the signal using conventional, manual techniques. It has been demonstrated that using manual and conventional approaches necessitates having prior problem-solving skills. This means that a characteristic retrieved from one issue or subject could not be desirable in another, reducing the classification accuracy. This problem has also been noted in earlier research. Furthermore, manual and conventional feature extraction techniques may raise the training process’s computational efficiency. Based on this, it is possible to conclude that manual and traditional feature extraction does not guarantee that the selected/extracted feature is best for the classifier. As a result, the examined techniques, which relied on laborious manual processes and conventional approaches, need to offer high reliability for automatically separating truth from falsehood. (B) It can be said that the EEG datasets used in previous research are only based on visual stimulation and are not based on questions and answers from the participants. To find the way to the practical field of the present research, it is necessary to design a more comprehensive database that records the signal based on auditory and speech stimuli so that it can be used in lie detection systems based on EEG signals.

The proposed method in this study for automatically distinguishing truth from falsehood is based on feature learning on EEG minimal channels. It combines deep graph convolutional and type 2 fuzzy networks to overcome the challenges above while demonstrating high reliability in practice. The contribution of this study can be summarized as follows:Providing an automatic lie detection system based on EEG signals with an accuracy of more than 95%.Collecting a standard database based on sentence questions and answers for the first time among previous research.Providing an automatic algorithm that uses a deep learning approach and type 2 fuzzy networks without needing a feature selection/extraction block diagram.The proposed model was evaluated in noisy environments, achieving accuracy above 90% in a wide range of different SNRs.

The rest of the article is organized as follows:

Section 2 examines the algorithms used in this study. Section 3 describes this research’s proposed method, which includes data registration, architectural design, etc. Section 4 presents the simulation results and compares the present study with algorithms and recent research. Finally, Section 5 is related to the conclusion.

## 2. Materials and Methods

This section begins with a description of the database for a lie detection system. Following that, the mathematical background of graph convolutional networks will be investigated.

### 2.1. General Model of Generative Adversarial Networks (GANs)

In recent years, GANs have gained significant attention as a vital subfield of deep learning. In 2014, J. Goodfellow and colleagues introduced these networks [18]. In machine learning, GANs handle unsupervised learning tasks. Two models that automatically identify and pick up patterns in the input data are part of these networks. We refer to these two models as discriminator and generator. To analyze, record, and duplicate changes in the dataset, the discriminator and the generator compete with one another. New samples that can be sensibly selected from the original dataset can be produced using GANs. The discriminator is trained using fictitious data produced by the generator. The generator gains the ability to generate usable data. Negative training samples are those that are produced for the discriminator. The generator creates a sample by using a fixed-length random noise vector as input. The generator’s primary objective is to deceive the discriminator into assigning the correct title to its output. Real data and fake data produced by the generator are separated by the discriminator. There are two distinct sources of training data for the discriminator. During training, the generator creates fake samples, which the discriminator uses as negative samples, while real data samples are used as positive samples.

In mathematical terms, the following equation is minimized in GAN networks during the training phase:(1)$$\log(1-D(G_{\left(Z\right)}))\;\begin{array}{ll} \frac{\min\max}{GD}V(G,D)= & E_{x}-P_{data}[\log D(x)]\\ & +E_{pz(z)}[\log(1-D(G(Z))] \end{array}$$In the above equation, the discriminator (D) must be obtained in such a way that it is possible to distinguish real and artificial data from each other. The equation introduced above cannot be solved in a closed form and requires repeated algorithms. Also, to avoid the problem of overfitting the data, for every *k* optimization of function *D*, generator function (*G*) is also optimized once [18].

### 2.2. General Model of Graph Convolutional Network

In 2016, Michael Deferard and colleagues initially put out the fundamental concept of the GCN. These researchers have applied signal processing to graphs and graph spectral theory for the first time, allowing for the derivation of convolutional functions and the use of convolutional networks in the setting of graph theory. Particularly significant in graph theory are the adjacency and degree matrices. An adjacency matrix is used to link each vertex in the graph. Moreover, the degree matrix may be obtained by having the adjacency matrix. The diagonal elements of this matrix, which is a diagonal matrix, are equal to the sum of the edges connecting to the appropriate vertex of the matrix. The degree matrix can be represented as $D\in R^{N\times N}$ and the graph matrix as $W\in R^{N\times N}$, where the *i*-th diagonal element of the degree matrix is defined as follows [19]:(2)$$D_{ii}=\sum_{i}W_{ij}$$The Laplacian matrix can also be defined in the form of the following relation:(3)$$L=D-W\in R^{N\times N}$$
(4)$$L=U\Lambda U^{T}$$According to the above relation, as it is known, the subtraction of the degree matrices and the adjacency matrix forms the Laplacian matrix. This matrix is used to calculate graph basis functions. Graph basis functions can be obtained using Singular Value Decomposition (SVD) in the Laplacian matrix. Also, the Laplacian matrix can be defined by considering the matrix of eigenvectors and the matrix of singular values in relation (5). According to Equation (5), the eigenvector matrix’s columns correspond to the Laplacian matrix’s eigenvectors. Fourier transform is also possible based on these eigenvectors, and Fourier bases can be defined by having diagonal eigenvalues including $\Lambda =diag([\lambda_{0,\ldots ,}\lambda_{N-1}])$ in the form of the following relationship:(5)$$U=[u_{0},\ldots ,u_{N-1}]\in R^{N\times N}$$For better understanding, the Fourier transform and inverse Fourier transform of a signal $q\in R^{N}$ can be defined in relations (7) and (8), respectively:(6)$$\hat{q}=U^{T}q$$
(7)$$q=UU^{T}q=U\hat{q}$$According to Equation (7), $\hat{q}$ represents the Fourier transform of the graph. Also, based on Equation (8), the feature vector for a signal such as q with Fourier bases and Fourier transform of the graph is possible. The graph convolution operator can also be calculated by having the convolution of two signals in the graph domain by the Fourier transform of each signal. For better understanding, the convolution of two signals *z* and *y* along with the operator ${*}_{g}$ is defined as the following relationship:(8)$$z{*}_{g}=U((U^{T}z)⊙(U^{T}y))$$In the above relation, g() filter function describes a graph convolution operator in combination with neural networks. According to the above relation, *z* is the version filtered by g(L):(9)y=g(L)zBy placing the Laplacian matrix and decomposing it into singular values and eigenvectors, graph convolution can be defined as follows [20]:(10)$$\begin{array}{l} y=g(L)z=Ug(\Lambda )U^{T}z\\ =U(g(\Lambda ))⊙(U^{T}z)\\ =U(U^{T}(U_{g}(\Lambda )))⊙(U^{T}z)\\ =z{*}_{g}(Ug(\Lambda )) \end{array}$$

### 2.3. General Model of Type 2 Fuzzy (TF-2)

Professor Zadeh introduced type 2 fuzzy (TF-2) sets in 1975 as a means of problem-solving and developing type 1 fuzzy (TF-1). Membership functions in TF-2 systems have membership degrees, setting them apart from TF-1 systems. TF-2 sets can withstand a wide range of uncertainties, including noise. These systems are helpful in designing control systems and predicting uncertain time series. However, these functions can also be used as activation functions in deep learning networks. As is well known, activation functions in deep learning networks have a significant impact on learning. The activation functions commonly used in deep learning networks include ReLU and Leaky-ReLU. These functions help to solve the gradient removal problem and improve the performance of deep learning networks. The main weakness of these functions is that their input and output relationships are nonlinear [21].

According to the introduced ability of TF-2 systems in this study, these sets have been used instead of ReLU and Leaky-ReLU activation functions in deep learning networks to deal with various uncertainties such as the nonlinearity of relationships between input and output, as well as to solve the noise effect. As stated above, the functions of these sets in deep learning networks can be defined as follows:(11)$$f\left(\sigma ;\gamma \right)=\left\{\begin{array}{l} P\sigma k(\sigma ),\;\;\;\mathrm{if}\;\sigma >\;0\\ N\sigma (-\sigma ),\;\mathrm{if}\;\sigma \leq 0 \end{array}\right.$$According to the above relationship, *k* can be defined as follows:(12)$$k\left(\sigma \right)=\frac{1}{2}\left(\frac{1}{\alpha +\sigma -\alpha \sigma }+\frac{-1+\alpha }{-1+\alpha \sigma }\right)$$When we have the mathematical derivatives of the introduced parameters, we can learn the γ=[α, P, N] parameters, which should be updated with each network iteration. The equation below demonstrates how to update these parameters:(13)$$\frac{\partial L}{\partial \gamma_{C}}=\sum_{j}\frac{\partial L}{\partial f_{c}(\sigma_{cj})}\frac{\partial f_{c}(\sigma_{cj})}{\partial \gamma_{c}}$$

The number of layers, the observation element, and the objective function in deep learning networks are related to parameters *c*, *j*, and *L*, respectively, according to the equation above. $\frac{\partial L}{\partial f_{c}(\sigma_{cj})}$ also represents the slope emanating from the deep layers, and the total slope is equal to the following equation:(14)$$\frac{\partial f_{c}(\sigma_{c})}{\partial a_{c}}=\left\{\begin{array}{l} \frac{p_{c}\sigma_{c}}{2}(\frac{1}{\alpha_{c}\sigma -1}+\frac{\sigma_{c}-1}{{\left(a_{c}+\sigma_{c}-\alpha_{c}\sigma_{c}\right)}^{2}}+\frac{\sigma_{c}(1-a_{c})}{{\left(a_{c}\sigma_{c}-1\right)}^{2}})\\ \mathrm{if}\;\sigma_{c}>0\\ -\frac{N_{c}\sigma_{c}}{2}(\frac{1}{\alpha_{c}\sigma +1}+\frac{\sigma_{c}+1}{{\left(a_{c}-\sigma_{c}+\alpha_{c}\sigma_{c}\right)}^{2}}+\frac{\sigma_{c}(1-a_{c})}{{\left(a_{c}\sigma_{c}+1\right)}^{2}}\\ \mathrm{if}\;\sigma_{c}\leq 0 \end{array}\right.$$
and we have:(15)$$\begin{array}{l} \frac{\partial f_{c}(\sigma_{c})}{\partial P_{C}}=\left\{\begin{array}{ll} \sigma_{c}k_{c}(\sigma_{c}), & \;\mathrm{if}\;\sigma_{c}>0\\ 0, & \;\mathrm{if}\;\sigma_{c}\leq 0\; \end{array}\right.\\ \frac{\partial f_{c}(\sigma_{c})}{\partial N_{C}}=\left\{\begin{array}{ll} 0, & \;\mathrm{if}\;\sigma_{c}>0\\ \sigma_{c}k_{c}(-\sigma_{c}), & \;\mathrm{if}\;\sigma_{c}\leq 0\; \end{array}\right. \end{array}$$$k_{c}(.)$ is also obtained from the parameters update law as follows:(16)$$\Delta \gamma =\rho \Delta \gamma +\xi \frac{\partial L}{\partial \gamma }$$This equation represents the amount of movement and the training rate, respectively.

Compared to the total number of weights in deep learning networks, the number of adjustable and learning parameters in TF-2 sets is only *3C* (where *C* is the number of hidden layers). This decreases the computational complexity significantly. To address different uncertainties, these sets have been used in this study’s graph convolutional networks instead of standard activation functions [21].

## 3. Proposed Model

This section will outline the suggested approach for creating an automatic system that detects lies using EEG signals. This part covers how to record a database, pre-processing of data, designed network architecture, optimization of designed architecture parameters and how to allocate training and test data. The study’s suggested flowchart is graphically depicted in Figure 1.

Figure 1 depicts the collection of a standard database based on EEG signals classified as truth or lie. The data will then be pre-processed using steps such as notch filtering, Butterworth filtering, data enhancement, and normalization. Following that, for feature selection/extraction and classification, the proposed network architecture, which combines TF-2 sets and graph convolutional networks, will be utilized. Finally, the data will be classified into truth and lies.

### 3.1. Data Collection

In order to collect data, 20 people (10 men and 10 women) of average age (20 to 35) with no underlying ailment were requested to take the lie detection test. First, the volunteers are informed that they are participating in the experiment voluntarily and that they have the right to leave at any time if they are dissatisfied with the experimental processes. The Tabriz University Faculty of Electrical and Computer Science’s ethics committee issued the necessary permits for signal recording (IR.Tabriz.1399.2.1). The subjects were asked two days before the trial not to consume caffeinated or energy drinks for 48 h. They were also urged to bathe before the test and avoid applying hair conditioners.

The Open BCI device recorded EEG signals according to the 10–20 standard. In this work, the data are recorded at a sampling frequency of 500 Hz, and EEG is measured with 16 channels of silver chloride. Also, EEG signals were recorded in bipolar form. To record the signal, channels A1 and A2 were used as references, with impedance matching set to less than 8 KΩ.

After receiving informed consent from the individuals, they were asked to answer questions in two separate scenarios. The questions included first and last names, father’s and mother’s names, places of education, birth, domicile, and national identification numbers. In the first scenario, participants are required to answer questions while EEG data are recorded accurately. After capturing the signal from the first scenario, the subjects are instructed to answer the identical questions that were wrong in the second scenario. Then, after the completion of signal registration, the first and second scenarios are labeled true and false, respectively. Each scenario’s signal recording process took 30 s. Accordingly, there were 15,000 samples (30 s × 500 Hz) for each lie and truth class. To avoid EOG noise, participants were asked to close their eyes while answering the questions. An example of the signals recorded from two scenarios of truth and lie from the F_Z_ channel is shown in Figure 2. According to this figure, there is no significant visual difference between the two different labels, which indicates the necessity of designing an automatic lie detection system. Also, Figure 3 depicts one of the individuals during signal recording with the Open BCI device.

### 3.2. Pre-Processing of EEG Data

As is evident, the data must be cleansed before entering the proposed network. As a result, this subsection describes in detail the pre-processing performed on the registered database. The executed pre-processing consists of five steps: in the first phase, according to studies [9,13], only channels Fz, Cz, Pz, O1, and O2 were employed, while the remaining EEG channels were left out. Decreasing the quantity of EEG channels diminishes the computational intricacy of the algorithm. Consequently, this enhances the algorithm’s efficiency and enables the model’s implementation in real-time applications. The second stage was using a Notch filter [22] to remove the 50 Hz frequency of city electricity from the data. In the third phase, a 2nd-order Butterworth filter [23] was applied to the data in the frequency range of 0.05 to 60 Hz to remove the participants’ random movements from the recordings. In the fourth step, GAN networks were utilized to increase the amount of recorded data and train the proposed network more effectively. The GAN network trains two subnetworks simultaneously: generator and discriminator. The generating network generates a 1×7500 dimensional signal from a 100-dimensional vector with a uniform distribution. This network’s five 1D-convolutional layers are being tested through trial and error. The layers’ diameters are 512, 1024, 2048, 4096, and 7500, respectively. Each layer employs batch normalization, whereas the network activation function is Leaky-ReLU. The network’s learning rate and number of iterations are 0.0001 and 200, respectively. The discriminant network accepts an 1×7500 dimensional vector as input and decides on the output (whether the signal is real or not). Furthermore, this network is made up of five dense fully connected layers. After employing this network, the data dimensions grew from 7500 to 10,000. In the fifth stage, the data are normalized between 0 and 1 [24] to aid network training.

### 3.3. Graph Design

A proximity matrix is generated after determining the functional connectivity of EEG channels. This can be accomplished by evaluating the correlation between the channels and showing the results as an EEG channel connection matrix. A threshold is specified for the connectivity matrix’s sparse approximation to remove the network adjacency matrix. The produced graph is fed into the suggested model, which selects/extracts and classifies features.

### 3.4. Customized Architecture

This subsection presents a proprietary network architecture for automatic lie detector detection. After using the dropout layer, the input is transmitted to six graph convolutional layers activated by TF-2. The dynamic information included in EEG signals is extracted using graph convolutional layers. After passing through batch normalization, the data will be triggered again using the TF-2 function. Following this phase, a dropout layer is added to prevent overfitting. Finally, the output is a flattening layer divided into two classes of truth and falsehood utilizing the ultimately linked layer and the Softmax activator. Figure 4 illustrates the described design graphically. In the customized design based on the convolutional graph, the number of graph nodes equals the number of channels considered. Thus, in the first convolution layer, each vertex receives 10,000 samples. Table 1 shows that the coefficients of S_1_, S_2_, S_3_, S_4_, S_5_, S_6_, and S6 represent each layer’s Chebi Sheff polynomial expansion [25] and differ between them. The dimensionality reduction in the layers of the proposed network is shown in Figure 5.

### 3.5. Training, Validation, and Test Series

The trial-and-error method determined the appropriate architecture for the proposed network. Table 2 shows the selected ideal parameters, such as the number of layers, layer type, optimization algorithms, filters, etc.

Data for training, validation, and test sets are similarly allocated randomly, with dimensions of 70%, 20%, and 10%, respectively.

## 4. Experimental Results

This part will show the suggested model’s outcomes. The proposed architecture was designed using the Python programming language, and the data preparation simulations were carried out in the MATLAB 2019a environment. The Google Colab ver 2024 Premium edition with a GPU t60 and 64 GB of RAM also produced the findings.

This research evaluated the results based on standard criteria such as accuracy, precision, sensitivity, and specificity. The evaluation of the above formulas can be defined as follows:(17)$$accuracy:\frac{TP+TN}{TP+TN+FP+FN}$$
(18)$$precision:\frac{TP}{TP+FP}$$
(19)$$sensitivity:\frac{TP}{TP+FN}$$
(20)$$specificity:\frac{TN}{TN+FP}$$According to the above relationships, *TP*, *TN*, *FN*, and *FP* represent the true positive, true negative, false negative, and false positive ratio, respectively.

This section has three subsections. The first subsection displays the optimization findings for the network architecture to visually demonstrate that the architecture considered for the current application is ideal. The second subsection shows the outcomes of the suggested model for the automated detection of lie detectors. The third and last portion compares the results with contemporary algorithms and research, one by one.

### 4.1. Architecture Optimization Results

The outcomes of the suggested network’s optimization are shown in this subsection. For this reason, Figure 6 demonstrates that the proposed model’s selection of six graph convolutional layers was appropriate for computation and efficiency. This chart shows that adding layers with a number higher than six increases computing efficiency while maintaining nearly stable accuracy in the network. Furthermore, we have considered polynomial coefficients in several ways when designing the suggested architecture; the outcomes are shown in Figure 7. This chart shows that the network performs best when the coefficients of *S*_1_-*S*_5_ = 1 are taken into account.

### 4.2. Results of Simulation

Figure 8 depicts the accuracy and error of the proposed network for automatic detection of lie detectors using fuzzy sets (proposed model), ReLU, and Leaky ReLU activation functions. As previously stated, 200 repetitions are considered for the proposed model (network), with stability beginning at 192 repeats. The network error has decreased after iteration 192. The significance of adopting TF-2 sets is demonstrated in this figure. Table 3 displays many evaluation criteria for distinguishing lies and facts, such as accuracy, precision, sensitivity, specificity, and the kappa coefficient. As it is known, all of the obtained values exceed 95%. Figure 9 depicts the confusion matrix and the receiver operating characteristic curve (ROC) plot analysis. According to this image, as the confusion matrix indicates, just two samples are incorrectly recognized in the suggested model, demonstrating the model’s perfect performance. Furthermore, the ROC diagram shows that the classification results are between 0.9 and 1 on the left side. Figure 10 displays the TSNE graph for raw EEG data and the FC layer. Based on the given figure, it is evident that the examples from two classes, truth and false, were combined in the raw data state. However, after inputting into the proposed network, the samples were successfully segregated into true and false classes in the final layer (fully connected layer). This indicates that the network has demonstrated high effectiveness in accurately classifying the two courses of truth and lies.

As is well known, EEG signals have a low SNR, and random movements of participants, such as blinking, might impair classification accuracy. As a result, the used model should have strong noise resistance. This study combined graph convolutional networks with TF2 to prevent a drop in classification accuracy due to noise. Gaussian white noise with a normal distribution was injected into the data at various SNR levels to demonstrate the model’s efficiency. Figure 11 depicts the performance of TF2 (proposed model) when compared to ReLU and Leaky-ReLU activation functions. As previously stated, the performance of the suggested model when using TF-2 functions can be more resistant to external noises than ReLU and Leaky ReLU activation functions.

### 4.3. Comparison with Previous Algorithms and Studies

This subsection will compare the proposed model’s performance to other recent one-on-one research.

Table 4 compares existing research and their methods with the proposed model. As is evident, the proposed technique outperforms recent investigations. So, the accuracy of the proposed model is 98.2%. However, the highest values of this coefficient for the [6] and [12] studies are 96 and 96.45%, respectively. The highest accuracy achieved among the studies is related to [14], which is around 98%. However, this study uses feature selection/extraction and manual classification. As mentioned in the Introduction, manual methods are unsuitable for real-time applications because they cause computational complexity.

None of the research employed a reference database to classify data. As a result, a one-on-one comparison with these studies appears unfair. As a result, we simulated our registered database using recently developed conventional methods and compared the results to our model. For this objective, pre-trained AlexNet [26], ResNet60 [27], and InceptionV3 [28] networks were compared to the proposed model. The results are shown in Figure 12. As shown in the figure, the proposed algorithm converged to the ideal value faster. Furthermore, as is well known, the suggested model has the highest level of accuracy when compared to other networks.

Despite its promising results, this research, like earlier ones, has limitations. This work utilized GAN networks to augment the data and prevent the model from overfitting during training. The size of the registered database can be enhanced in the future, eliminating the need to add data artificially. In addition, wet electrodes were utilized to record the signal in this work, which can be explored in future investigations of dry electrode performance.

## 5. Conclusions

This study presents a fully automatic model for detecting truth from lies using EEG signals. This study’s proposed model is based on the combination of TF-2 sets and graph convolutional networks and is end-to-end, eliminating the need for a feature selection/extraction block diagram. In this study, a standard database of EEG signals from 20 subjects was collected. The classification findings revealed that the suggested model has a high accuracy of 98%, which is quite promising compared to previous studies. The algorithm’s promising performance allows the suggested model to be applied in various lie detection applications. In future research, we intend to use the proposed algorithm as a real-time model for lie detection using minimal channels of EEG signals.

## Figures and Tables

**Figure 1 sensors-24-03598-f001:**
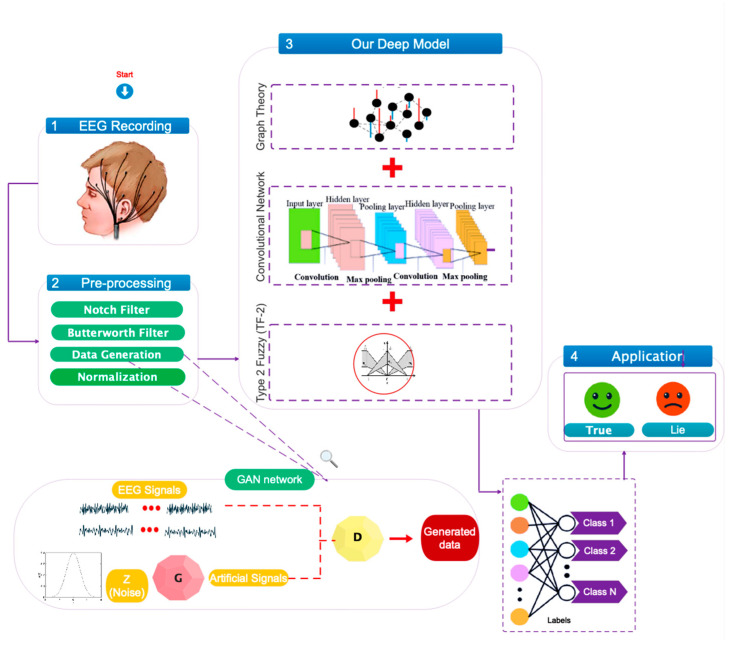
The main chart depicts automatic lie detection using EEG signals via a combination of TF-2 sets and deep graph convolutional networks.

**Figure 2 sensors-24-03598-f002:**
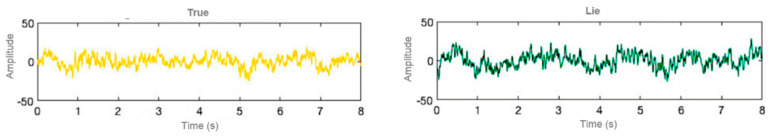
EEG signal was recorded for two labels of truth and lie from the F_Z_ channel.

**Figure 3 sensors-24-03598-f003:**
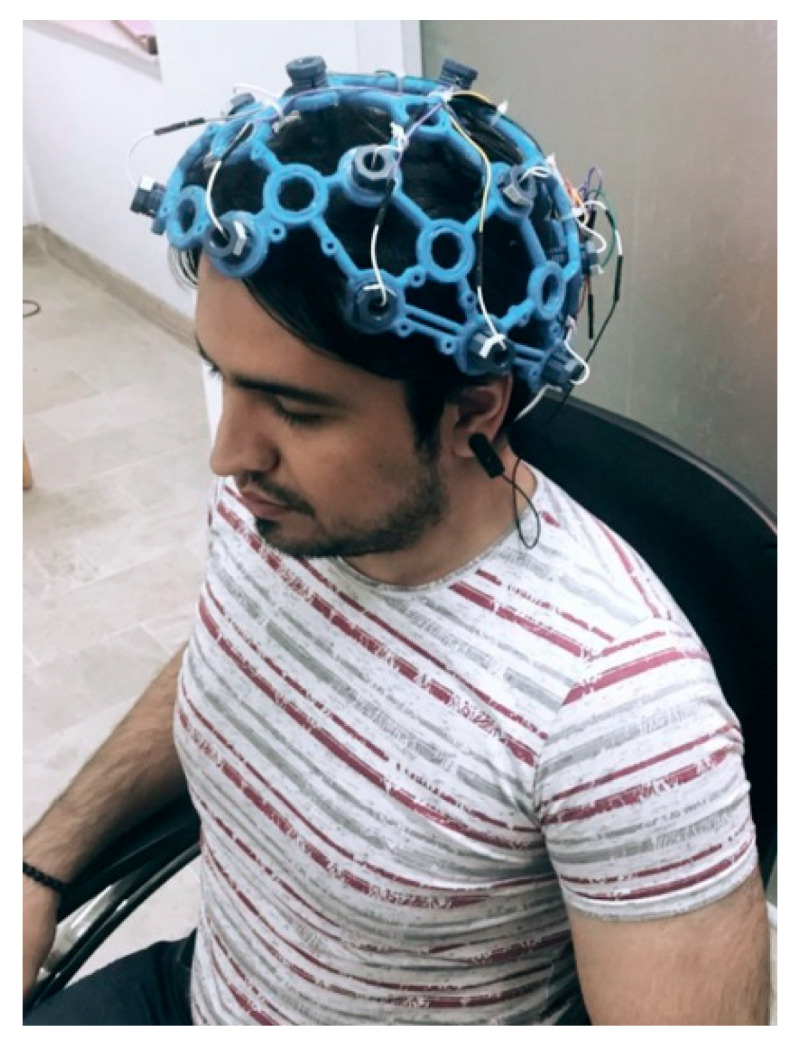
Recording of EEG signals using Open BCI modules on one of the participants.

**Figure 4 sensors-24-03598-f004:**
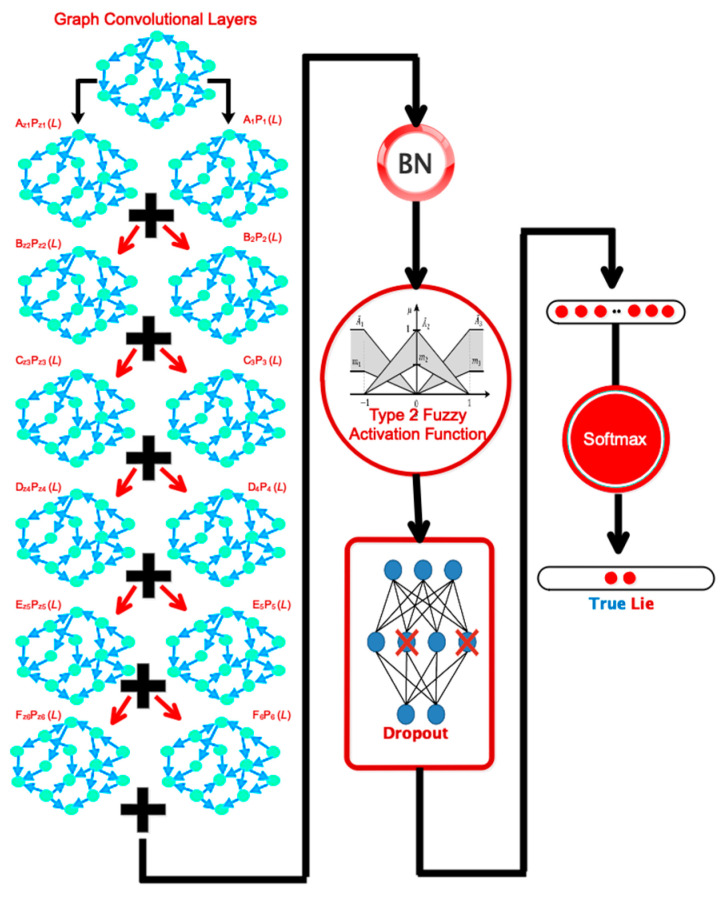
The architecture of the proposed model for automatic lie detection.

**Figure 5 sensors-24-03598-f005:**
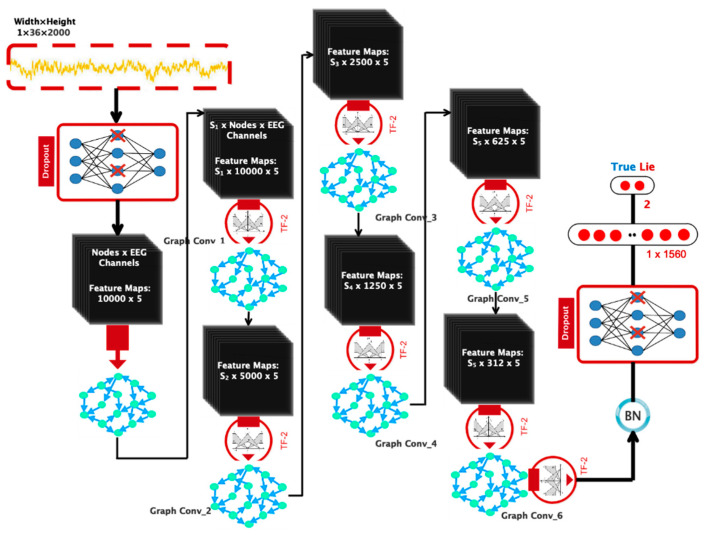
A suggested deep network architecture that includes layer details.

**Figure 6 sensors-24-03598-f006:**
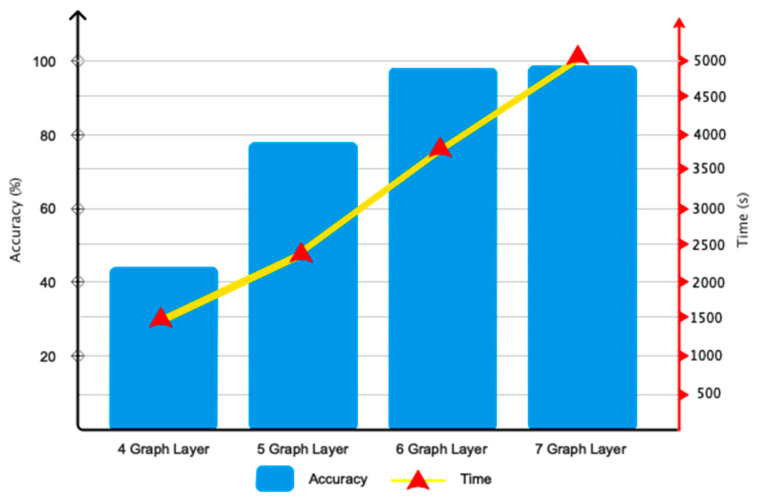
Selective performance of proposed network layers.

**Figure 7 sensors-24-03598-f007:**
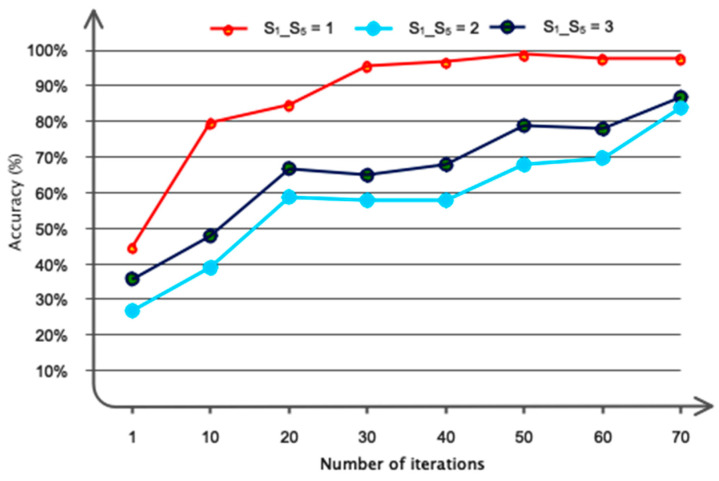
Different polynomial coefficients were examined in the graph convolutional architecture.

**Figure 8 sensors-24-03598-f008:**
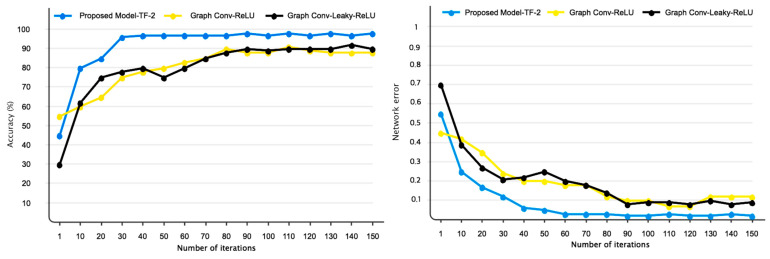
The accuracy and error of the proposed model were compared to different activation functions.

**Figure 9 sensors-24-03598-f009:**
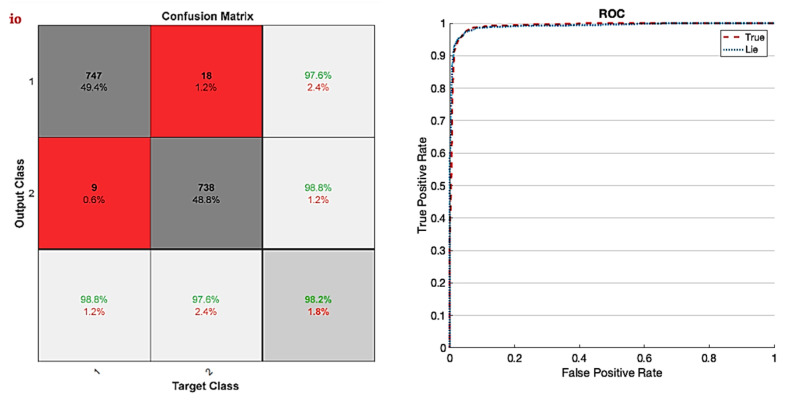
Confusion matrix (**Left**) with ROC curve analysis (**Right**).

**Figure 10 sensors-24-03598-f010:**
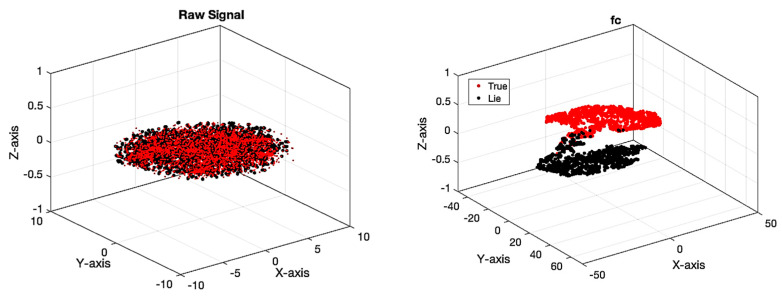
Samples of two classes of truth and falsehood for raw data and the fully connected network layer.

**Figure 11 sensors-24-03598-f011:**
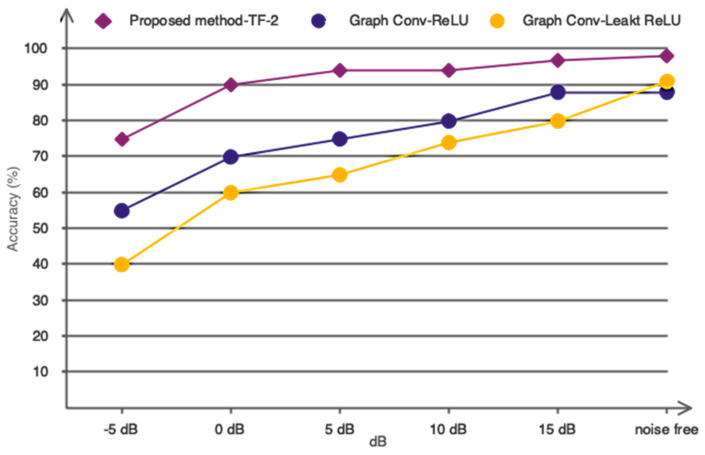
Comparison of network performance with different activation functions in noisy environments.

**Figure 12 sensors-24-03598-f012:**
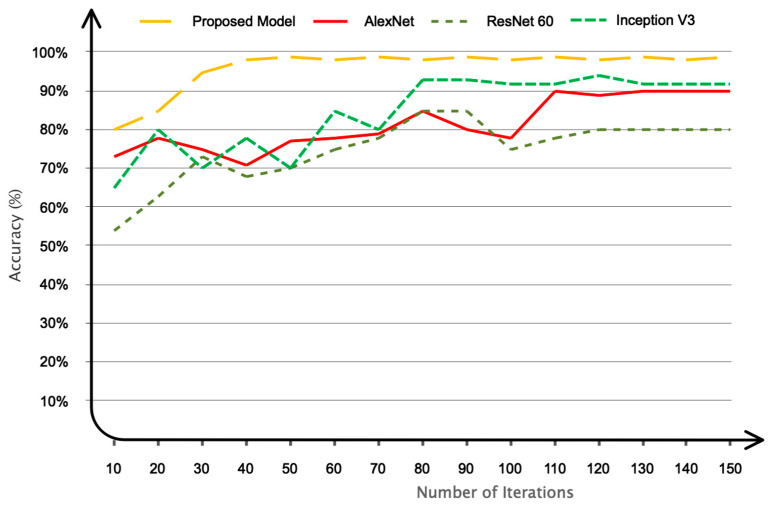
The proposed network’s performance in comparison to other networks.

**Table 1 sensors-24-03598-t001:** The number of filters, stride size, and architectural details of the customized CNN model.

Layer	Shape of Weight Tensor	Shape of Bias	Number of Parameters
**Graph Conv1**	(*S*_1_, 10,000, 10,000)	10,000	100,000,000 × *S*_1_ + 10,000
**Graph Conv2**	(*S*_2_, 10,000, 5000)	5000	50,000,000 × *S*_2_ + 5000
**Graph Conv3**	(*S*_3_, 5000, 2500)	2500	12,500,000 × *S*_3_ + 2500
**Graph Conv4**	(*S*_4_, 2500, 1250)	1250	3,125,000 × *S*_4_ + 1250
**Graph Conv5**	(*S*_5_, 1250, 625)	625	781,250 × *S*_5_ + 625
**Graph Conv6**	(*S*_6_, 625, 312)	312	195,000 × *S*_6_ + 312
**Flattening Layer**	624	2	1248

**Table 2 sensors-24-03598-t002:** The suggested network architecture’s ideal parameters were chosen.

Parameters	Values	Optimal Value
Batch Size in GAN	4, 6, 8, 10, 12	10
Optimizer in GAN	Adam, SGD, Adamax	SGD
Number of CNN Layers	3, 4, 5	4
Learning Rate in GAN	0.1, 0.01, 0.001, 0.0001	0.0001
Number of Graph Conv Layers	2, 3, 4, 5, 6, 7	6
Batch Size in GCN	8, 16, 32	16
Batch normalization	ReLU, Leaky-ReLU, TF-2	TF-2
Learning Rate in GCN	0.1, 0.01, 0.001, 0.0001, 0.00001	0.001
Dropout Rate	0.1, 0.2, 0.3	0.1
Weight of optimizer	$4\times 10^{-3},4\times 10^{-4},4\times 10^{-5},4\times 10^{-6},4\times 10^{-7}$	$4\times 10^{-6}$
Error function	MSE, Cross Entropy	Cross Entropy
Optimizer in GCN	Adam, SGD, Adadelta, Adamax	Adadelta

**Table 3 sensors-24-03598-t003:** The performance of the proposed network is based on different evaluation indices.

Measurement Index	Performance (%)
**Accuracy**	98.2
**Sensitivity**	98.2
**Precision**	98.1
**Specificity**	98.3
**Kappa coefficient**	0.93

**Table 4 sensors-24-03598-t004:** Comparison of the proposed model with recent studies.

Research	The Method Used	ACC (%)
Abootalebi et al. [9]	P300 Waves	86
Amir et al. [10]	Classical Features	80
Mohammad et al. [11]	Brain Waves	79
Gao et al. [12]	SVM	96
Simbolon et al. [13]	ERP	83
Saini et al. [14]	SVM	98
Yohan et al. [15]	ANN	86
Bagel et al. [16]	CNN	84
Dodia et al. [17]	FFT-Hand Crafted Features	88
Kang et al. [4]	ICA + FCN	88.5
Boddu et al. [6]	PSO + SVM	96.45
Our Model	GAN + Fuzzy Graph Convolution	98.2

## Data Availability

The data are private and the University Ethics Committee does not allow public access to the data.

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
