# Peer review of "An Automatic Lie Detection Model Using EEG Signals Based on the Combination of Type 2 Fuzzy Sets and Deep Graph Convolutional Networks"

_sensors, 2024, doi:10.3390/s24113598_

Round 1
Reviewer 1 Report
Comments and Suggestions for Authors
The paper presents a new model for automatic lie detection using EEG signals. The proposed model combines deep graph convolutional networks and type 2 fuzzy sets for feature selection/extraction and automatic classification. The results show that the suggested model is effective in discriminating between truth and lies, even in noisy settings, with a classification accuracy higher than 90%. However, the paper still has the following problems:
The resolution of the figures needs to be substantially improved. Clear figures help improve the quality of your paper and make it easier for readers to read.
Expression of contribution is insufficient. The expression of contribution is not good enough. It only states what the article does, but does not clearly express the innovation of the article. It should be refined and rewritten.
The abstract may be vague or unclear and fail to convey the key information of the study. It is recommended that you revisit the abstract and ensure that you use concise and clear language so that readers can quickly understand the purpose, methods and key findings of the study.
Could you provide more details on the dataset used in the study, such as the number of samples, the distribution of truth and lie labels, and any demographic information about the participants?
How did you determine the optimal number of layers and the polynomial coefficients for the graph convolutional network? Did you perform any experiments or analysis to justify these choices?
Have you compared the performance of the proposed model with other state-of-the-art lie detection methods? If not, could you discuss the potential advantages and limitations of your approach compared to existing techniques?
Conclusion writing is inadequate. There is no good way to summarize the full text and express your opinions.
English needs to be improved through small revisions and polishing.
Comments on the Quality of English Language
English needs to be improved through small revisions and polishing.
Author Response
Reviewer#1:
Recommendation: Minor Revision
Comments:
The paper presents a new model for automatic lie detection using EEG signals. The proposed model combines deep graph convolutional networks and type 2 fuzzy sets for feature selection/extraction and automatic classification. The results show that the suggested model is effective in discriminating between truth and lies, even in noisy settings, with a classification accuracy higher than 90%. However, the paper still has the following problems:
- ⎫ While thanking the esteemed reviewer for a thorough review of the manuscript version. We, the authors of the article, believe that your suggestions have been very useful and effective in improving the scientific version of the manuscript. We carefully answered all the questions and suggestions of the esteemed reviewer and added them to the manuscript version.
- 1. The resolution of the figures needs to be substantially improved. Clear figures help improve the quality of your paper and make it easier for readers to read.
- ⎫ The manuscript is revised based on this comment. According to the reviewer, all of the figures in the manuscript have been improved and replaced with newer ones.
- 2. Expression of contribution is insufficient. The expression of contribution is not good enough. It only states what the article does, but does not clearly express the innovation of the article. It should be refined and rewritten.
- ⎫ The manuscript is revised based on this comment. According to the reviewer's opinion, The contribution of this study can be summarized as follows:
- 1. Providing an automatic lie detection system based on EEG signals with an accuracy of more than 95%.
- 2. Collecting a standard database based on sentence questions and answers for the first time among previous researches.
- 3. Providing an automatic algorithm that uses a deep learning approach and type 2 fuzzy networks without needing a feature selection/extraction block diagram.
- 4. The proposed model was evaluated in noisy environments, achieving accuracy above 90% in a wide range of different SNRs.
which are highlighted on page 4 lines 158-167.
- 3. The abstract may be vague or unclear and fail to convey the key information of the study. It is recommended that you revisit the abstract and ensure that you use concise and clear language so that readers can quickly understand the purpose, methods and key findings of the study.
- ⎫ The manuscript is revised based on this comment. According to the reviewer's opinion, the abstract has been reviewed and modified for better and clearer readability.
“In recent decades, many different governmental and non-governmental organizations have used lie detection for various purposes, including ensuring the honesty of criminal confessions. As a result, this diagnosis is evaluated with a polygraph machine. However, the polygraph instrument has limitations and needs to be more reliable. This study introduces a new model for detecting lies using electroencephalogram (EEG) signals. An EEG database of 20 study participants was created to accomplish this goal. This study also used a 6-layer graph convolutional network and type 2 fuzzy (TF-2) sets for feature selection/extraction and automatic classification. The classification results show that the proposed deep model effectively distinguishes between truths and lies. As a result, even in a noisy environment (SNR=0 dB), the classification accuracy remains above 90%. The proposed strategy outperforms current research and algorithms. Its superior performance makes it suitable for a wide range of practical applications.”
Which are highlighted on page 1, lines 13-23.
- 4. Could you provide more details on the dataset used in the study, such as the number of samples, the distribution of truth and lie labels, and any demographic information about the participants?
- ⎫ The manuscript is revised based on this comment. All information related to the database is provided in section 3.1. Based on the opinion of the honorable judge, the number of samples in each class has been added to the database section as follows:
“In order to collect data, 20 people (10 men and 10 women) of average age (20 to 35) with no underlying ailment were requested to take the lie detection test. First, the volun-teers are informed that they are participating in the experiment voluntarily and that they have the right to leave at any time if they are dissatisfied with the experimental processes. Tabriz University Faculty of Electrical and Computer Science's ethics committee issued the necessary permits for signal recording (IR.Tabriz.1399.2.1). The subjects were asked two days before the trial not to consume caffeinated or energy drinks for 48 hours. They were also urged to bathe before the test and avoid applying hair conditioners.
The Open BCI device recorded EEG signals according to the 10-20 standard. In this work, the data is recorded at a sampling frequency of 500 Hz, and EEG is measured with 16 channels of silver chloride. Also, EEG signals were recorded in bipolar form. To record the signal, channels A1 and A2 were used as references, with impedance matching set to less than 8 KΩ.
After receiving informed consent from the individuals, they were asked to answer questions in two separate scenarios. The questions included first and last names, father's and mother's names, places of education, birth, domicile, and national identification numbers. In the first scenario, participants are required to answer questions while EEG data are recorded accurately. After capturing the signal from the first scenario, the subjects are instructed to answer the identical questions that were wrong in the second scenario. Then, after the completion of signal registration, the first and second scenarios are labeled true and false, respectively. Each scenario's signal recording process took 30s. According-ly, there were 15000 samples (30s x 500 Hz) for each lie and truth class.”
which are highlighted on page 8 line 306.
- 5. How did you determine the optimal number of layers and the polynomial coefficients for the graph convolutional network? Did you perform any experiments or analysis to justify these choices?
- ⎫ The trial-and-error method was used to determine the appropriate architecture for the proposed network. Table 2 shows the selected ideal parameters, such as the number of layers, layer type, optimization algorithms, filters, and so on. Figure 6 demonstrates that the suggested model's selection of six graph convolutional layers was appropriate in terms of computation and efficiency. This chart shows that adding layers with a number higher than six increases computing efficiency while maintaining nearly stable accuracy in the network.
Table 2. The suggested network architecture's ideal parameters were chosen.
Figure 6. Selective performance of proposed network layers.
which are highlighted on pages 11-12 line 368 and line 392.
- 6. Have you compared the performance of the proposed model with other state-of-the-art lie detection methods? If not, could you discuss the potential advantages and limitations of your approach compared to existing techniques?
- ⎫ According to the opinion of the respected referee, we have compared our research with other studies and methods as follows:
Table 4 compares existing research and their methods with the proposed model. As is obvious, the proposed technique outperforms recent investigations. So that the accuracy of the proposed model is 98.2%; However, the highest value of this coefficient for [6] and [12] studies is 96 and 96.45%, respectively. The highest accuracy achieved among the studies is related to [14], which is around 98%. However, this study uses feature selection/extraction and manual classification. As mentioned in the introduction section, the use of manual methods is not suitable for real-time applications; because they cause computational complexity.
Table 4. Comparison of the proposed model with recent studies.
| Research | The method used | ACC (%) | 
| Abootalebi et al. [9] | P300 Waves | 86 | 
| Amir et al. [10] | Classical Features | 80 | 
| Mohammad et al. [11] | Brain Waves | 79 | 
| Gao et al. [12] | SVM | 96 | 
| Simbolon et al. [13] | ERP | 83 | 
| Saini et al. [14] | SVM | 98 | 
| Yohan et al. [15] | ANN | 86 | 
| Bagel et al. [16] | CNN | 84 | 
| Dodia et al. [17] | FFT-Hand Crafted Features | 88 | 
| Kang et al. [4] | ICA+FCN | 88.5 | 
| Boddu et al. [6] | PSO+SVM | 96.45 | 
| Our Model | GAN + Fuzzy Graph Convolution | 98.2 | 
None of the research employed a reference database to classify data. As a result, a one-on-one comparison with these studies does not appear fair. As a result, we simulated our registered database using recently developed conventional methods and compared the results to our model. For this objective, pre-trained AlexNet [26], ResNet60 [27], and InceptionV3 [28] networks were compared to the proposed model. The results are shown in Figure 12. As shown in the figure, the proposed algorithm converged to the ideal value faster. Furthermore, as is well known, the suggested model has the highest level of accuracy when compared to other networks.
Figure 12. The proposed network's performance in comparison to other networks.
Also, the limitations of the research are as follows:
Despite its promising results, this study, like earlier ones, has limitations. In this study, GAN networks were utilized to augment the data and prevent the model from overfitting during training. The size of the registered database can be enhanced in the future, eliminating the need to add data artificially. In addition, in this study, wet electrodes were utilized to record the signal, which can be explored in future investigations of dry electrode performance.
- 7. Conclusion writing is inadequate. There is no good way to summarize the full text and express your opinions.
- ⎫ The manuscript is revised based on this comment. According to the opinion of the respected referee, the conclusion of the article was modified and rewritten as follows:
“This study presents a fully automatic model for detecting truth from lies using EEG signals. This study's proposed model is based on the combination of TF-2 sets and graph convolutional networks and is end-to-end, eliminating the need for a feature selection/extraction block diagram. In this study, a standard database of EEG signals from 20 subjects was collected. The classification findings revealed that the suggested model has a high accuracy of 98%, which is quite promising compared to previous studies. The algorithm's promising performance allows the suggested model to be applied in various lie detection applications. In future research, we intend to use the proposed algorithm as a real-time model for lie detection using minimal channels of EEG signals.”
Which are highlighted on page 16, line 471.
- 8. English needs to be improved through small revisions and polishing.
- ⎫ The manuscript is revised based on this comment. The grammar of the manuscript was double-checked and spelling and grammatical errors in the manuscript were corrected.

Reviewer 2 Report
Comments and Suggestions for Authors
In this manuscript, the authors have presented a new model for detecting lies using electroencephalogram (EEG) signals, and verified by experiments. Overall, this manuscript is well conducted and clearly written. But some issues should be addressed before publication.
(1) There is no explanation on how the specific evaluation value is calculated. So it is suggested to provide calculation formulas for Measurement indexes. For Measurement indexes, such as accuracy, precision, there is no explanation on how the specific evaluation value is calculated.
(2) What is the main limitation of the suggested model? Can you explain the main drawback or limitation of the proposed method compared to the existing classic methods?
(3) It is suggested to improve the figure resolutions, such as Figs. 6-8, 11-12. And the font size of these Figures should be larger.
(4) The format of some references needs to be adjusted necessarily.
So in this work, it should be introduced which signal recording mode is used to acquire EEG signals? Bipolar montage or unipolar montage?
(5) Compared to Type-1 Fuzzy Sets, what are the advantages of the Type-2 Fuzzy Sets used in this work? How is the membership interval determined in TF-2?
Comments on the Quality of English Language
Good English expression.
Author Response
Reviewer#2:
Recommendation: Minor Revision
Comments:
In this manuscript, the authors have presented a new model for detecting lies using electroencephalogram (EEG) signals, and verified by experiments. Overall, this manuscript is well conducted and clearly written. But some issues should be addressed before publication.
- ⎫ While thanking the esteemed reviewer for a thorough review of the manuscript version. We, the authors of the article, believe that your suggestions have been very useful and effective in improving the scientific version of the manuscript. We carefully answered all the questions and suggestions of the esteemed reviewer and added them to the manuscript version.
- 1. There is no explanation on how the specific evaluation value is calculated. So it is suggested to provide calculation formulas for Measurement indexes. For Measurement indexes, such as accuracy, precision, there is no explanation on how the specific evaluation value is calculated.
- ⎫ The manuscript is revised based on this comment. Yes, the opinion of the honorable judge is absolutely correct. According to the opinion of the respected reviewer, we have added various evaluation formulas such as accuracy, precision, sensitivity and specificity to the manuscript. “This research evaluated the results based on standard criteria such as accuracy, precision, sensitivity, and specificity. The evaluation of the above formulas can be defined as follows:
| 
 | (17) | 
| 
 | (18) | 
| 
 | (19) | 
| 
 | (20) | 
According to the above relationships, TP, TN, FN, and FP represent the true positive, true negative, false negative, and false positive ratio, respectively.”
which are highlighted on page 11 line 382.
- 2. What is the main limitation of the suggested model? Can you explain the main drawback or limitation of the proposed method compared to the existing classic methods?
- ⎫ Despite its promising results, this study, like earlier ones, has limitations. In this study, GAN networks were utilized to augment the data and prevent the model from overfitting during training. The size of the registered database can be enhanced in the future, eliminating the need to add data artificially. In addition, in this study, wet electrodes were utilized to record the signal, which can be explored in future investigations of dry electrode performance.
- 3. It is suggested to improve the figure resolutions, such as Figs. 6-8, 11-12. And the font size of these Figures should be larger.
- ⎫ The manuscript is revised based on this comment. According to the reviewer, all of the figures in the manuscript have been improved and replaced with newer ones.
- 4. The format of some references needs to be adjusted necessarily.
- ⎫ The manuscript is revised based on this comment. The format of all the references have been modified and organized based on the MDPI format.
which are highlighted on page 17.
- 5. So in this work, it should be introduced which signal recording mode is used to acquire EEG signals? Bipolar montage or unipolar montage?
- ⎫ The manuscript is revised based on this comment. We have added a bipolar record type to the database record section in section 3.1. which are highlighted on page 8, line 294.
- 6. Compared to Type-1 Fuzzy Sets, what are the advantages of the Type-2 Fuzzy Sets used in this work? How is the membership interval determined in TF-2?
- ⎫ The manuscript is revised based on this comment. With respect to the opinion of the reviewer, the proposed network architecture is organized in a trial and error manner, and accordingly, the momentum and gamma parameters in the proposed architecture are set to 0.8.
Unlike type 1 fuzzy systems, membership functions in type 2 fuzzy systems have fuzzy membership degrees. The use of type 2 fuzzy membership functions significantly increases the ability of fuzzy systems in facing uncertainties (including structural and measurement noise) compared to normal fuzzy systems (type 1 fuzzy functions).

Reviewer 3 Report
Comments and Suggestions for Authors
sensors-2986761: “An automatic lie detection model using EEG signals based on the combination of type 2 fuzzy sets and deep graph convolutional networks ”.
In this study, the authors develop a fully automatic system based on EEG signals for distinguishing truth from falsehood with high accuracy and speed.
This lie detector system overcomes the limitations of conventional “polygraph” approaches in this field and potentially builds a bridge between ensuring security and crime prevention and its control.
Suggested model with a row of iterations for performance optimization with use of both a small number of EEG signals and created comprehensive database based on speech and hearing stimuli is original in the subjected area.
Further improvements in this model might be associated with the enhancement of the registered database size.
In this study, all main questions posed have been addressed and the data obtained support the conclusions.
Minor remark:
1) in line 56 and below, “et al.” should be replaced by “and colleagues”;
2) in line, 72, “”...to 28 years.”;
3) in line 130, “...EEG channels were improved...” should be rewritten;
4) in line 179, “...of the database for ischemic stroke maps.” needs explanation;
5) Figure 1 needs more detailed description and bigger fonts;
6) in Methods, no explanation of what EEG channels were used in this study and why;
7) Figure 8, the ordinate title should be “Accuracy (%) without “%” on the axis;
8) Figure 10 needs more detailed description;
9) in line 464, “This work...”;
10) in line 494, year of publication should be added.
Comments on the Quality of English Language
Minor editing of English language required
Author Response
Reviewer#3:
Recommendation: Minor Revision
Comments:
sensors-2986761: “An automatic lie detection model using EEG signals based on the combination of type 2 fuzzy sets and deep graph convolutional networks ”.
In this study, the authors develop a fully automatic system based on EEG signals for distinguishing truth from falsehood with high accuracy and speed.
- ⎫ While thanking the esteemed reviewer for a thorough review of the manuscript version. We, the authors of the article, believe that your suggestions have been very useful and effective in improving the scientific version of the manuscript. We carefully answered all the questions and suggestions of the esteemed reviewer and added them to the manuscript version.
- 1. This lie detector system overcomes the limitations of conventional “polygraph” approaches in this field and potentially builds a bridge between ensuring security and crime prevention and its control.
- ⎫ Yes, the opinion of the honorable judge is absolutely correct. The conventional device used in lie detection, which is called polygraph, has fundamental limitations and is accompanied by errors. The proposed model based on EEG signals can distinguish lies from truth with less error.
- 2. Suggested model with a row of iterations for performance optimization with use of both a small number of EEG signals and created comprehensive database based on speech and hearing stimuli is original in the subjected area.
- ⎫ The manuscript is revised based on this comment. Yes, it is absolutely true. In the proposed model, a comprehensive database based on EEG signals has been collected in two classes, truths and lies.
- 3. It Further improvements in this model might be associated with the enhancement of the registered database size.
- ⎫ Yes, for this purpose, we used GAN, which consists of two sub-networks including generator and discriminator. The GAN network trains two sub-networks at the same time: generator and discriminator. The generating network generates a dimensional signal from a 100-dimensional vector with a uniform distribution. This network's five 1D-convolutional layers are being tested through trial and error. The layers' diameters are 512, 1024, 2048, 4096 and 7500, respectively. Each layer employs batch normalization, whereas the network activation function is Leaky-ReLU. The network's learning rate and number of iterations are set at 0.0001 and 200, respectively. The discriminant network accepts a 1 dimensional vector as input and decides on the output (whether the signal is real or not). Furthermore, this network is made up of five dense fully connected layers. After employing this net-work, the data dimensions grew from 7500 to 10000.
- 4. In this study, all main questions posed have been addressed and the data obtained support the conclusions.
- ⎫ . Thank you for the reviewer's opinion, yes, we have tried to compare our proposed model with other researches and previous methods. Compared to previous studies, the proposed model has higher accuracy and has the ability to be implemented in real-time systems.
“Table 4 compares existing research and their methods with the proposed model. As is obvious, the proposed technique outperforms recent investigations. So that the accuracy of the proposed model is 98.2%; However, the highest value of this coefficient for [6] and [12] studies is 96 and 96.45%, respectively. The highest accuracy achieved among the studies is related to [14], which is around 98%. However, this study uses feature selection/extraction and manual classification. As mentioned in the introduction section, the use of manual methods is not suitable for real-time applications; because they cause computational complexity.
Table 4. Comparison of the proposed model with recent studies.
| Research | The method used | ACC (%) | 
| Abootalebi et al. [9] | P300 Waves | 86 | 
| Amir et al. [10] | Classical Features | 80 | 
| Mohammad et al. [11] | Brain Waves | 79 | 
| Gao et al. [12] | SVM | 96 | 
| Simbolon et al. [13] | ERP | 83 | 
| Saini et al. [14] | SVM | 98 | 
| Yohan et al. [15] | ANN | 86 | 
| Bagel et al. [16] | CNN | 84 | 
| Dodia et al. [17] | FFT-Hand Crafted Features | 88 | 
| Kang et al. [4] | ICA+FCN | 88.5 | 
| Boddu et al. [6] | PSO+SVM | 96.45 | 
| Our Model | GAN + Fuzzy Graph Convolution | 98.2 | 
None of the research employed a reference database to classify data. As a result, a one-on-one comparison with these studies does not appear fair. As a result, we simulated our registered database using recently developed conventional methods and compared the results to our model. For this objective, pre-trained AlexNet [26], ResNet60 [27], and InceptionV3 [28] networks were compared to the proposed model. The results are shown in Figure 12. As shown in the figure, the proposed algorithm converged to the ideal value faster. Furthermore, as is well known, the suggested model has the highest level of accuracy when compared to other networks.
Figure 12. The proposed network's performance in comparison to other networks.”
- 5. Minor remark:
- 1) in line 56 and below, “et al.” should be replaced by “and colleagues”;
- ⎫ The manuscript is revised based on this comment which are highlighted on introduction section.
- 2) in line, 72, “”...to 28 years.”:
- ⎫ The manuscript is revised based on this comment.
- 3) in line 130, “...EEG channels were improved...” should be rewritten;
- ⎫ The manuscript is revised based on this comment.
- 4) in line 179, “...of the database for ischemic stroke maps.” needs explanation;
- ⎫ The manuscript is revised based on this comment which are highlighted on section 2, line 175.
- 5) Figure 1 needs more detailed description and bigger fonts;
- ⎫ The manuscript is revised based on this comment. The study's suggested flowchart is graphically depicted in Figure 1.
Figure 1. The main chart depicts automatic lie detection using EEG signals via a combination of TF-2 sets and deep graph convolutional networks.
- ⎫ Figure 1 depicts the collection of a standard database based on EEG signals classified as truth or lie. The data will then be pre-processed using steps such as notch filtering, Butterworth filtering, data enhancement, and normalization. Following that, for feature selection/extraction and classification, the proposed network architecture, which combines TF-2 sets and graph convolutional networks, will be utilized. Finally, the data will be classified into two classes: truth and lies.
which are highlighted on section 3, line 275.
- 6) in Methods, no explanation of what EEG channels were used in this study and why;
- ⎫ The manuscript is revised based on this comment.
“In the first phase, according to studies [9, 13], only channels Fz, Cz, Pz, O1, and O2 were employed, while the remaining EEG channels were left out. Decreasing the quantity of EEG channels diminishes the computational intricacy of the algorithm. Consequently, this enhances the algorithm’s efficiency and enables the model's implementation in re-al-time applications”
which are highlighted on page 9, line 321.
- 7) Figure 8, the ordinate title should be “Accuracy (%) without “%” on the axis;
- ⎫ The manuscript is revised based on this comment which are highlighted on page 13, line 423.
Figure 8. The accuracy and error of the proposed model were compared to different activation functions.
- 8) Figure 10 needs more detailed description;
- ⎫ The manuscript is revised based on this comment.
“Figure 10 displays the TSNE graph for raw EEG data and the FC layer. Based on the given figure, it is evident that the examples from two classes, truth and false, were combined in the raw data state. However, after inputting into the proposed network, the samples were successfully segregated into true and false classes in the final layer (fully connected layer). This indicates that the network has demonstrated high effectiveness in accurately classifying the two courses of truth and lies.”
which are highlighted on page 13, line 416.
- 9) in line 464, “This work...”;
- ⎫ The manuscript is revised based on this comment.
10) in line 494, year of publication should be added.
- ⎫ The manuscript is revised based on this comment which are highlighted on page 17.

Round 2
Reviewer 1 Report
Comments and Suggestions for Authors
The author responded point-to-point to the reviewers' suggestions and revised the paper. I have no more comments.
Comments on the Quality of English LanguageThe author responded point-to-point to the reviewers' suggestions and revised the paper. I have no more comments.